# Characterization and Discrimination of Commercial Portuguese Beers Based on Phenolic Composition and Antioxidant Capacity

**DOI:** 10.3390/foods10051144

**Published:** 2021-05-20

**Authors:** Irene Gouvinhas, Cristiana Breda, Ana Isabel Barros

**Affiliations:** Center for the Research and Technology of Agro-Environmental and Biological Sciences (CITAB)/Inov4Agro (Institute for Innovation, Capacity Building and Sustainability of Agri-Food Production), University of Trás-of-Montes and Alto Douro (UTAD), 5000-801 Vila Real, Portugal; igouvinhas@utad.pt (I.G.); Cristiana.breda@hotmail.com (C.B.)

**Keywords:** commercial beers, phenolic composition, High-Performance Liquid Chromatography–Diode Array Detector, antioxidant capacity, correlation

## Abstract

Beer has been highly appreciated due to its phenolic composition and antioxidant capacity conjugated with its low alcohol content. Although some studies exist regarding the phenolic composition and antioxidant capacities of beers, there are no studies related to the determination of these parameters in the most commonly consumed commercial beers in Portugal. The phenolic composition and antioxidant capacity of 23 Portuguese commercial beers of different styles and types were studied for the first time. The total phenolic content, ortho-diphenols, and flavonoids ranged between 0.15 ± 0.01 and 0.82 ± 0.07 g Gallic Acid (GA) L^−1^; 0.07 ± 0.02 and 1.80 ± 0.09 g GA L^−1^, and 0.02 ± 0.00 and 0.15 ± 0.02 g Catechin (CAT) L^−1^, respectively. An accurate quantitative phenolic analysis was also performed, and the compound identified with a higher amount was gallic acid, followed by syringic acid. Concerning flavonoids, gallo-catechin was the most abundant compound identified (from 21.44 ± 2.87 and 144.00 ± 10.93 μg mL^−1^). A significant correlation between ortho-diphenols and the antiradical capacity (ABTS and DPPH) was found, the latter being negatively correlated. Flavonoids content was also positively correlated with total phenols and antiradical capacity determined by the ABTS assay. These results evidence that phenolic composition is affected by several factors inherent to beers, namely ingredients, fermentation type, and brewing process.

## 1. Introduction

Beer is the most widely consumed alcoholic beverage in the world and the third most popular drink after water and tea [1,2], a complex alcoholic beverage made from barley (malt), hop, water, and brewer’s yeast, rich in nutrients, such as carbohydrates, amino acids, vitamins, and minerals, as well as non-nutrient components, such as phenolic acids and flavonoids, mainly derived from the added malt (70–80%) and hops (20–30%) [3,4]. However, during beer production, these secondary metabolites undergo chemical modifications, forming new molecules which will influence either the yield or the final characteristics of beers. Furthermore, the metabolic activity of microbes on raw materials is responsible for the astringency, flavor, taste, aroma and body of beers, mainly influenced by the phenolic compounds [5].

Yeast plays a key role in the sensory development of beer, the fermentation being carried out by a limited variety of yeast strains, most of which are from the genus *Saccharomyces*. Ale beers, more popular in northern countries, are normally fermented at warmer temperatures (20 ± 4 °C) for shorter time periods with the top strain, *Saccharomyces cerevisiae*. On the other hand, Lager beers are produced with strains of *Saccharomyces calsbergensis*, fermented at cooler temperatures (from 4 to 12 °C) during several weeks (4–12 weeks), being the most consumed beers all over the world [6]. Other beer classifications are related to the brewing process, which results in alcoholic and non-alcoholic beers, draft beers, beers with flavors, or pale and brown beers, depending also on cultural practices, geographic origin, and ingredients’ availability [7].

Much has been written about the favorable impact of moderate consumption of red wine on the human body. However, critical assessment of the literature indicates that beer appears to be just as beneficial in countering diseases, such as coronary heart disease and cancer [8]. In fact, the rich constitution of phenolic compounds and, consequently, the high antioxidant capacity revealed by beers, are the main factors responsible for these biological properties. Furthermore, these secondary metabolites are also related to the increase in the shelf-life of beers once they intervene in the brewing process, due to the delay, retardation, and prevention of the oxidation processes [4,9]. The most important phenolic compounds found in commercial beers are xantho-humol (originated from hop), malt phenolics (for example: phenolic acids, flavanols and flavonols), amino phenolic compounds, pro-anthocyanidins, and tannins [10,11]. All these compounds are responsible for important characteristics of beers, namely color, aroma and flavor, colloidal stability and the ability to interact with proteins, causing turbidity [5]. However, the phenolic concentration and profile of beers depend on several factors, not only on beer style, but also on the composition of the raw materials employed during the brewing process, which also influences the composition of these compounds [12].

Recently, some studies have demonstrated the enrichment of beers with bioactive compounds, namely carotenoids and polyphenols, by the addition of fruits (such as cherry, raspberry, peach, apple, quince fruits, etc.) during the fermentation process, contributing new organoleptic and functional characteristics to beers and their alcoholic content, as well as to potential health benefits when consumed moderately [2,13,14].

Although some studies exist regarding the phenolic composition and antioxidant capacities of beers, to the best of our knowledge, no studies relating to the determination of these parameters in the most commonly consumed commercial beers in Portugal have been performed. In this sense, the aims of this work were to study and to characterize some of the most common Portuguese beers available on the market, by the assessment of their phenolics content and antioxidant capacity. Additionally, a deep statistical analysis was conducted through principal component analysis (PCA), a Pearson correlation test, and a dendogram to evaluate regularity and differences among samples, specifically to distinguish differences in the phenolic content and antioxidant capacity of beer samples according to their styles. Thus, the purpose of this investigation was to provide a further insight into polyphenolic compounds present in beers of different types and styles which can be found on the Portuguese market, since no similar studies have been conducted in Portugal.

## 2. Materials and Methods

### 2.1. Chemicals

Folin-Ciocalteu’s reagent, acetic acid, 3,4,5-trihydroxybenzoic acid (gallic acid) both extra pure (>99%), sodium hydroxide, and potassium hydroxide were purchased from Panreac (Panreac Química S.L.U., Barcelona, Spain). Aluminum chloride, sodium nitrate, and sodium carbonate, all extra pure (>99%), and methanol were acquired from Merck (Merck, Darmstadt, Germany). Sodium molybdate (99.5%) was purchased from Chem-Lab (Chem-Lab N.V., Zedelgem, Belgium). The compounds 2,2-azino-bis (3-ethylbenzothiazoline-6-sulphonic acid) diammonium salt (ABTS^•+^), 2,2-diphenyl-1-picrylhidrazyl radical (DPPH^•^), potassium phosphate, acetonitrile, and standard compounds (all mass spectrometric grades) were obtained from Sigma-Aldrich (Steinheim, Germany). Additionally, 6-hydroxy-2,5,7,8-tetramethylchroman-2-carboxylic acid (Trolox) was purchased from Fluka Chemika (Neu-Ulm, Switzerland). Ultrapure water was obtained using a Millipore water purification system.

### 2.2. Sampling

The present work was carried out on three bottles of twenty-three Portuguese commercial beers purchased in supermarkets (Vila Real, Portugal), including dark beers, pale beers, non-alcoholic beers, and flavored beers. The brand names were omitted and represented by number codes, as summarized in Table 1, in which some characteristics were described as reported on the bottles. Just three of the beers were classified as ale according to the beer label. The remaining samples were classified as lager beers.

Before the analysis, beers were firstly degassed by mechanical agitation during 24 h [15]. Then, the samples were stored at 4 °C and analyzed within 48 h.

### 2.3. Phenolic Composition

For the determination of the phenolic content, namely total phenols, flavonoids, and ortho-diphenols, spectrophotometric assays were assessed, according to the methodologies previously reported [16].

Regarding the content of total phenolics in beers, the Folin-Ciocalteu method was employed, using gallic acid as standard, the results expressed as mg of gallic acid per liter (g GA L^−1^).

The content of ortho-diphenols in beers was determined by adding Na_2_MoO_4_ (50 g L^−1^) to the samples, appropriately diluted. The absorbance was recorded at 375 nm, and quantified using gallic acid as standard. Results were expressed as g GA L^−1^.

For the assessment of flavonoid content in beer samples, the aluminum complex method was performed, using catechin as standard. Results were expressed as mg of catechin per liter (g CAT L^−1^).

### 2.4. Antioxidant Capacity

The free radical scavenging activity was determined by DPPH and ABTS spectrophotometric methods adapted to a microscale, according to the procedure described by Leal et al. (2020), by measuring the variation in absorbance at 520 nm after 15 min of reaction of the phenolic compounds with DPPH^•^, adding 190 μL of the DPPH solution (8.87 mM) to 10 μL of the sample, and at 734 nm after 30 min for ABTS^•+^ (20.00 mM), mixing 188 μL of ABTS and 12 μL of the sample [17]. All the assays were also performed using 96-well micro plates (Nunc) and an Infinite M200 microplate reader (Tecan). The results were expressed in mmol Trolox per liter of beer sample (mmol Trolox L^−1^).

### 2.5. Chromatographic Determination of Phenolic Compounds

The polyphenolic profile of the beer samples was assessed by Reverse Phase—High Performance Liquid Chromatography—Diode Array Detector (RP-HPLC-DAD) (Thermo Finnigan Surveyor, San Diego, CA, USA) according to Gouvinhas et al., 2020, with some modifications [17]. In short, chromatographic analyses were carried out on a C18 column (250 × 4.6 mm, 5 µm particle size; ACE, Aberdeen, Scotland). Chromatographic separation was performed using distilled water/formic acid (99.9:0.1, *v*/*v*) (Solvent A) and acetonitrile/formic acid (99.9:0.1, *v*/*v*) (Solvent B) in the linear gradient scheme (t in min; %B): (0; 10%), (40; 26%), (70; 65%), (71; 100%), (75; 100%), (76; 10%), and (85, 10%). The injection volume and the flow rate were 20 µL and 1.0 mL min^−1^, respectively. Chromatograms were recorded in the range 200–600 nm range and analyzed at 280 and 330 nm. The equipment consisted of a LC pump (SRVYR-LPUMP) (Thermo Finnigan Surveyor, San Diego, CA, USA), an auto-sampler (SRVYR-AS) (Thermo Finnigan Surveyor, San Diego, CA, USA), and a photodiode array detector (SRVYR-PDA5) (Thermo Finnigan Surveyor, San Diego, CA, USA) in succession. Concentrations were expressed in micrograms per milliliter of sample (µg mL^−1^).

### 2.6. Statistical Analysis

All the assays (phenolic composition and antioxidant capacity) were performed using 96-well micro plates (Nunc, Roskilde, Denmark) and an Infinite M200 microplate reader (Tecan, Grödig, Austria). For all analyses, three replicates (*n* = 3) of each sample were assessed. The results are presented as mean (*n* = 3) ± standard deviation (SD). The data obtained were subjected to variance analysis (ANOVA) and a multiple range test (Tukey’s test) for a *p* value < 0.05, using IBM SPSS statistics 21.0 software (SPSS Inc., Chicago, IL, USA) [17]. A correlation analysis was also performed between variables using the Pearson test. In order to evaluate regularity and differences among samples, namely to distinguish differences in the phenolic content and antioxidant capacity of beer samples, a multivariate analysis was performed using PCA (Principal Component Analysis) [18]. PCA transforms the large number of potentially correlated factors (the data) into a set of values of uncorrelated variables (principal components, PCs), and thus minimizes the size of the data set. The number of PCs selected was the minimum sufficient to reproduce the data within a required degree of accuracy. The first procedure employed to perform the PCA analysis was the normalization of the dataset. The used values were manually normalized to 0–1 range, taking into account the maximum value obtained from each assay [18]. Afterwards, a covariance matrix from the standardized data was created, which presented five columns (variables) and 23 rows (observations), followed by the calculation of the PCs (eigenvectors) and their corresponding eigenvalues. The components were then sorted in descending order of their eigenvalues and further plotted in a graph. This statistical test was performed using the software Origin Pro 9.1. A dendrogram analysis was also carried out in order to evaluate the similarity between the samples, using the values obtained for the principal components, the results being projected as Euclidean distance. The values obtained for the principal components were also applied in a cluster analysis.

## 3. Results and Discussion

### 3.1. Total Phenols, Ortho-Diphenols and Flavonoid Contents of Portuguese Beers

The results of the phenolic content of Portuguese beers regarding total phenols, ortho-diphenols, and flavonoids are presented in Table 2.

As can be observed, the sample 2-Black revealed the highest concentration of total phenols, with 0.824 ± 0.074 g GA L^−1^, being significantly different from all the other beers under study, with the exception of beer 1, both ale beers, thus subjected to a high fermentation process; in contrast, the beer with the lowest concentration was 23-Panache, with 0.153 ± 0.013 g GA L^−1^, significantly distinguishable from other beers.

Similar concentrations to those described above were obtained by Zhao et al. (2010), namely between 0.152 and 0.340 g GA L^−1^ for 34 commercial lager-type beers also using the Folin-Ciocalteau method [9]. However, other authors found significantly lower values of total phenols in laboratory, local Canadian and foreign commercial beers, namely between 0.04 and 0.14 g GA L^−1^ [15]. Oladokun et al. (2016) also analysed commercial lager beers in terms of total phenols, obtaining concentrations between 0.07 and 0.26 g GA L^−1^, much lower than those found in the present study [19]. In contrast, Nardini and Foddai (2020) investigated the content of total phenols of seven commercial special beers (produced with the addition of natural foods during the fermentation process, such as chestnut, honey, cocoa, green tea, etc.), obtaining significantly higher values than those obtained in the present study with values ranging between 0.464 and 1.026 g GA L^−1^, which are significantly lower than conventional beers analysed by the same authors (from 0.2734 and 0.446 g GA L^−1^) [20]. The results of total phenols content published by Marques et al. (2017), in a study of phenolic profile and antioxidant activity of different craft beer styles, ranged between 0.448 and 0.531 g GA L^−1^, which are similar to those obtained in the present study, despite being craft beers [21]. Also concerning craft beers, Pereira et al. (2020) analyzed the physicochemical composition of wheat craft beers with additional cashew peduncle and orange peel during the brewing process. In their study, the values obtained for total phenols ranged from 0.515 to 0.729 g GA L^−1^, which were lower than those obtained in our study [22]. The significant differences between the results of commercial beers and craft beers may be due to many variations during the brewing processes and the addition of unconventional raw materials, such as fruits or spices [23].

Regarding the content of ortho-diphenols (Table 2), the 7-White sample proved to be the beer with the highest concentration with 1.801 ± 0.087 g GA L^−1^, being significantly different from all the other beers studied (*p* < 0.05); in contrast, the beer with the lowest concentration of ortho-diphenols was 8-White beer with 0.074 ± 0.024 g GA L^−1^.

Analyzing the flavonoids, the results revealed that 2-Black beer again presented (just as for total phenols) the highest concentration, with a content of 0.151 ± 0.017 g CAT L^−1^; the beer with the lowest concentration was the 15-Lemon, with 0.020 ± 0.005 g CAT L^−1^. The results regarding flavonoids published by Nardini and Foddai (2020), in a study of the phenolic composition of special and conventional beers, ranged from 0.042 to 0.096 g CAT L^−1^ and from 0.027 to 0.063 g CAT L ^−1^ of beer, respectively, which are in the range of those determined in the present study [20].

In their study, Rahman et al. (2020) obtained a flavonoid content in the range of 0.008 and 0.05 g CAT L^−1^, much lower than those found in this study. Nardini and Garaguso (2020) also obtained higher values of phenolic compounds content (total phenols and flavonoids) in ale beers than in lager [13], which is in agreement with our study, namely concerning the sample 2-Dark, and other studies [24,25].

The differences found in total phenols and flavonoid content between values of the present study and those in the literature could be due to beer storage conditions, beer type, origin of raw ingredients, brewing techniques, fermentation type, and time during the industrial process [7].

### 3.2. Antioxidant Capacity of Portuguese Beers

The antioxidant capacity of the 23 beer samples was evaluated and the results are presented in Table 2. Regarding the evaluation of the anti-radicular capacity through the ABTS method, the results obtained revealed that 2-Black and 4-Black beers were the beers with the highest capacity (0.102 ± 0.002 mmol trolox L^−1^ and 0.107 ± 0.002 mmol trolox L^−1^), significantly different from all the other samples; in contrast, beers that showed the lowest anti-radicular capacity were the Lemon beers number 14, 15, and 16 (around 0.008 ± 0.001 mmol trolox L^−1^), not significantly different from beers 11 (White) and 23 (Panache).

According to Zhao et al. (2010) the values obtained for ABTS were in the range of 0.55 mmol trolox L^−1^ to 1.95 mmol trolox L^−1^, clearly above those obtained in the present study (0.008 and 0.107 mmol trolox L^−1^) [9]. Nardini & Foddai (2020) also investigated the antioxidant capacity of special and conventional beers. The values obtained for special beers ranged from 2.4 to 5.2 mM trolox L^−1^, which are below the values obtained in the conventional samples (1.5–2.6 mM Trolox L^−1^) [20]. Concerning craft beers, Pereira et al. (2020) also evaluated the antioxidant capacity of this kind of beer and obtained values (1.568–1.737 mmol trolox L^−1^) much higher than those obtained in the present study [22]. These differences may be explained by the different formulations during the production process of craft beers [12].

Concerning the DPPH methodology, the results obtained (Table 2) revealed the 10-White sample as the beer with the greatest anti-radicular capacity, with 0.038 ± 0.00 mmol trolox L^−1^, not significantly different from other samples. In contrast, 2-Black beer showed the lowest values, with 0.019 ± 0.002 mmol trolox L^−1^, not significantly different from beers 1 (Abbey), 3 (Black), 4 (Black), 5 (White), 6 (White), and 7 (White).

Comparing the results of the present study with those of Zhao et al. (2010), we found that these DPPH values were higher, ranging from 0.24 to 0.70 mmol trolox L^−1^ in lager-type beers. Rahman et al. (2020) also obtained higher DPPH values (between 0.25 and 0.71 mmol trolox L^−1^) to those of the present study (0.019–0.038 mmol trolox L^−1^) [15].

Tafulo et al. (2010) also determined the antioxidant capacity of 27 commercial beers, 18 of them Portuguese. This activity was determined by six different methods and three different standards. Despite the high antioxidant power found in these samples, the ORAC method presented the highest values for antioxidant capacity. Furthermore, among the several factors that can influence the antioxidant capacity of beers, it was found that the color and the method employed were the factors that most significantly affected this activity [26].

In the present study, the significant differences found in some samples can also be explained by the fact that they come from different manufacturers and, despite the similar base components used in beer production and process, there may be differences in terms of the quantities established by each manufacturer and the origin of each component.

The presence of phenolic compounds in beer can thus provide an antioxidant action, making it able to assist with some physiological disorders of the body, without worrying about the effects of alcohol, due to its low content.

### 3.3. Phenolic Profile of Portuguese Beers

The identification and quantification of phenolics was also performed by Reverse Phase–High-Performance Liquid Chromatography–Diode Array Detector (RP-HPLC-DAD) to obtain more detailed information about the phytochemical composition of these Portuguese beers. Table 3 presents the phenolic compounds identified according to the standards used, divided into phenolic acids and flavonoid compounds.

It was possible to observe that the most abundant compound (from non-flavonoids) present in almost all beer samples was gallic acid, followed by syringic acid. The highest concentration of gallic acid found was in 1-Abbey beer, with 95.80 ± 3.71 µg mL^−1^, while the highest concentration of syringic acid was for the 12-White non-alcoholic beer, with a concentration of 12.40 ± 1.62 µg mL^−1^.

Contrarily to the non-flavonoids previously described, other non-flavonoid compounds were detected in low concentrations in beer samples analyzed. In fact, protocatechuic and ferulic acids were only detected in two samples, namely in 2-Dark and 5-White beers, and 2-Dark and 15-Lemon beers, respectively, 2-Dark sample being the beer with the highest concentration for both compounds (28.43 ± 0.26 and 1.49 ± 0.06 µg mL^−1^, respectively). This is in agreement with some studies in the literature which quantified less phenolic compounds in lager beers than in ales, with lower concentrations of *p*-coumaric, syringic, and caffeic acids [13].

Other compounds were only found in one beer sample, namely hydroxybenzoic, caffeic, and vanillic acids in 2-Dark (12.33 ± 0.248 µg mL^−1^), 22-Red fruits (29.24 ± 3.64 µg mL^−1^), and 18-Bohemia wheat (1.332 ± 0.086 µg mL^−1^), respectively. Nardini et al. (2020) also found caffeic acid, among other compounds, in fruit beers, which are significantly improved in phenolic compounds comparatively to conventional beers.

Zhao et al. (2010) also identified some of these compounds in beer samples. Concerning gallic acid, the concentrations found in the present work were ten times higher than those obtained by these authors (10.39 ± 0.09 µg mL^−1^ and 1.81 ± 0.11 µg mL^−1^). Concerning protocatechuic acid, in the present study higher values than those obtained by the authors Zhao et al. (2010) were found, obtaining concentrations between 0.02 ± 0.02 µg mL^−1^ and 1.30 ± 0.05 µg mL^−1^. In contrast, the concentration in ferulic acid in the present study was less than that found by the same authors, who obtained values of 0.51 ± 0.03 µg mL^−1^ and 2.81 ± 0.04 µg mL^−1^ in their beer samples [9]. Finally, caffeic acid, which was only identified in one beer, was also found at higher concentrations comparatively to the concentrations in the study of Zhao et al. (2010) (between 0.12 ± 0.03 µg mL^−1^ and 1.22 ± 0.05 µg mL^−1^). In a different study, Nardini & Foddai (2020) compared the phenolic profile of conventional and special beers by HPLC. In a conventional beer some compounds were identified, namely ferulic acid, caffeic acid, sinapic acid, vanillic acid, *p*-coumaric acid and syringic acid, at levels of 10.27 to 21.66 μg mL^−1^, 1.61–5.99 μg mL^−1^, 2.19–4.80 μg mL^−1^, 2.30–4.65 μg mL^−1^, 0.77–2.77 μg mL^−1^ and 0–0.71 μg mL^−1^, respectively. These values were similar to those obtained in special beers, in which the levels of ferulic acid, caffeic acid, sinapic acid, vanillic acid, *p*-coumaric acid and syringic acid were in the ranges of 8.22–27.55 μg mL^−1^, 1.48 to 9.20 μg mL^−1^, 2.52–6.73 μg mL^−1^, 2.03–5.09 μg mL^−1^, 1.75–4.32 μg mL^−1^, and 0.67–1.42 μg mL^−1^, respectively [20]. Comparing these results with the present study, the values of vanillic and ferulic acids are much higher than those determined in the present work, in opposition to the concentration of caffeic and syringic acids which were much lower than those determined in this study. Marques et al. (2017), who evaluated four styles of craft beer, identified compounds, namely gallic acid, caffeic acid, *p*-coumaric acid and ferulic acid at levels in the ranges 0.33–1.71 μg mL^−1^, 3.95–9.05 μg mL^−1^, 0.12–0.39 μg mL^−1^, and 2.12–4.02 μg mL^−1^, respectively [21]. Although these authors studied craft beers, which are commonly enriched in bioactive compounds due to the several raw materials employed, in our study significantly higher values of gallic acid and caffeic acid were found. However, ferulic acid was present at a lower concentration in our study.

Concerning flavonoids (Table 3), gallo-catechin was the most abundant compound found in all the samples (from 21.44 ± 2.87 and 144.00 ± 10.93 µg mL^−1^), except for 2-Black beer in which it was not detected. Epicatechin and catechin were also found in high concentrations in almost all beers, with concentrations ranging from 3.28 ± 0.18 and 215.10 ± 22.58 µg mL^−1^ and from 1.41 ± 0.01 and 155.30 ± 2.10 µg mL^−1^, respectively.

The other compounds identified, including kaempferol derivatives, rutin, and quercetin, were present in a small number of samples, namely in Lemon (14 and 16) and Red fruits beers (22). Concerning kaempferol, this compound was found in eleven beer samples, and in higher concentration in dark and ale samples, namely 1-Abbey, 2-Dark, and 3-Dark (0.22 µg mL^−1^, on average).

In fact, several differences were found between the beer samples, due to several factors, such as fermentation type, origin, concentration, quality of ingredients, and the brewing process. Furthermore, the phenolic compounds present in beer come from hops and, in the majority, from barley malt, making the drink a source of polyphenols [27]. However, compounds derived from hops are easier to characterize than those from barley, since during the processing of the drink the latter can undergo changes, making the compounds difficult to characterize [28]. Despite this, during the malting and brewing process, phenolic compounds are also subject to changes due to extraction or enzymatic release, to heat-induced chemical reactions, or to precipitation with or adsorbance to hot and cold trub, stabilization agents and yeast cells [29].

Within the several stages that make up the beer production process, filtration emerges as one of the main factors responsible for the drastic reduction in the content of polyphenols present in the final matrix. The boiling stage also causes a series of changes in the must polyphenols’ composition, which is already complex, making it difficult to predict the fate of the polyphenols in this mixture. Such complexity is partly due to the ease of oxidation and polymerization of several phenolic compounds [30].

Thus, in barley grains, derivatives of hydroxybenzoic and hydroxycinnamic acids have been identified, such as *trans*-ferulic acid, found in greater quantity in the grain, followed by *p*-coumaric and vanillic acids. These acids are known to act as primary antioxidants in the reception of free radicals, interrupting the chain reaction, and are present in the aleurone layer and in the endosperm of the grain [31].

Generally, phenolic compounds are found in beer linked to other compounds, such as esters and glycosides, but it is also possible to find them in their free form, and some substances are more likely to be found in malt or hops. Derivatives of hydro-benzoic acids and hydroxinamic acids, such as ferulic, *p*-coumaric and caffeic acids (also identified in this work), are more commonly extracted from malt, while flavonols, chalcones and flavanones are essentially found from hops. Equally detectable, both in hops and in malt, are tannins derived from flavonols, catechins and procyanidins [32].

In addition to several dependent factors and the contribution to the aroma and color of the beers, the phenolic compounds are correlated with antioxidant capacity, improving the stability and, consequently, the shelf-life of beers.

### 3.4. Characterization of Portuguese Beers by PCA Analysis

Principal component analysis (PCA) represents one of the most widely used chemometric tools. PCA is an unsupervised technique, which reduces the dimensionality of the original data matrix while maintaining maximum variability and allows visualization of the original arrangement of the samples in an n-dimensional space through the information maintained, allowing the relationship between variables and observations to be studied, as well as the recognition of the data structure. The explanation of the differences in the samples is given through the factors obtained from the generalized correlation matrix of the data sets and, at the same time, allows the determination of which variables contribute most to this differentiation (Figure 1A).

PCA was applied to evaluate data on the main phenolic content determined, namely ortho-diphenols, total phenols, flavonoids and antioxidant capacity (ABTS and DPPH) (Figure 1B). The first main component was able to explain 56.87% of the total variance and the second explained 26.32%, totalling 83.19%.

The simple dispersion graph (Figure 2A) suggests the location of the beers in relation to phenolic content and antioxidant capacity. It was possible to observe the formation of five groups. The group represented by the dark square was the one with the highest DPPH values, the group represented by plus symbols was the one with the highest values for ortho-diphenols, the group represented by cardinals showed higher values for the flavonoids and total phenols, and, lastly, the group represented by asterisks, showed the highest values for antioxidant capacity, namely ABTS.

The similarity of the samples was evaluated using hierarchical analysis of clusters and five groups were suggested (Figure 2B), which they corroborate the results found by the PCA. The means for each response variable were compared statistically. Through the separation of the groups, it was possible to observe that in cluster 1 there are 2-Black and 4-Black beers, which are similar in anti-radicular capacity by ABTS assay. In cluster 2, the samples 1-Abbey and 3-Black were included, because they are in the quadrant of anti-radicular capacity by ABTS and content of *ortho*-diphenols. In cluster 3, the beers 5-White, 6-White and 7-White were grouped, since they have identical content of ortho-diphenols, and hence they were grouped in the same quadrant. In cluster 4 were the samples 8-White, 12-White without alcohol, 13-Black without alcohol, 17-90 years (edition), 18-Bohemia wheat, 19-Bohemia pure malt, 20-Original bohemia, 21-Bohemia IPA, and 22-Red fruits due to the common content of flavonoids, total phenols, and anti-radicular capacity values for DPPH. Finally, in cluster 5 were 9-White, 10-White, 11-White, 14-Lemon without alcohol, 15-Lemon, 16-lemon, and 23-Panache. These beers presented common characteristics in terms of ortho-diphenols and the same anti-radicular capacity for DPPH.

Through Pearson’s statistical correlation analysis (*p* ≤ 0.05) it was possible to observe, through the results obtained that the analyzes performed obtained a significant correlation with each other (Table 4). These correlations are also shown in Table 4.

There is a significant correlation between ortho-diphenols and the anti-radicular capacity of ABTS and DPPH, the latter negatively correlated.

In the case of total phenols, the correlation was significant for flavonoids and anti-radicular capacity for ABTS and DPPH, the latter also being negatively correlated. For flavonoids, the correlation was significant for total phenols and for anti-radicular capacity for ABTS.

Concerning the anti-radicular capacity, this correlated with ortho-diphenols, total phenols and flavonoids, correlating negatively once again with anti-radicular capacity by DPPH.

## 4. Conclusions

Beer can be considered a good source of polyphenols, which can come from both malt and hops. Due to its antioxidant capacity and low alcohol content, beer has been extensively studied in its capacity to reduce the risk of coronary heart disease. In this study, an accurate qualitative and quantitative determination of phenolic compounds by chromatographic and spectrophotometric methods has been performed on commercial Portuguese beers. The HPLC-DAD analyses allowed the determination of seven phenolic acids and eleven flavonoids in 23 commercial beers.

The phenolic profile was characterized by high contents of gallic and syringic acids, kaempferol, gallo-catechin and epicatechin, and low contents of vanillic, ferulic, and caffeic acids, quercetin, and rutin. High correlations have been found between some phenolic contents and antioxidant capacity determined by ABTS and DPPH methods.

## Figures and Tables

**Figure 1 foods-10-01144-f001:**
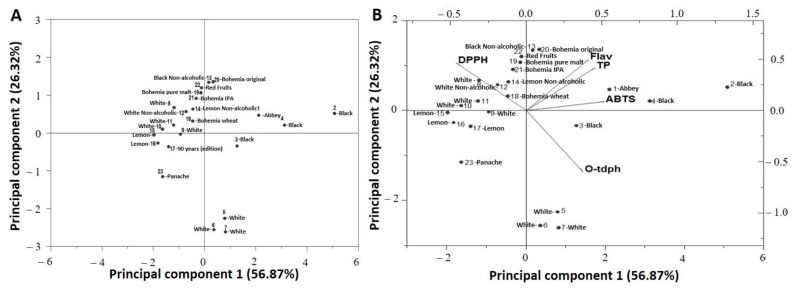
(**A**) Simple dispersion graph (Principal component 1 × Principal component 2) over the main sources of beer variability; (**B**) Projection of the values obtained for PCs 1 and 2 for the different samples. The weights of each variable for each of the factors are represented by the position of the abbreviations in the quadrants.

**Figure 2 foods-10-01144-f002:**
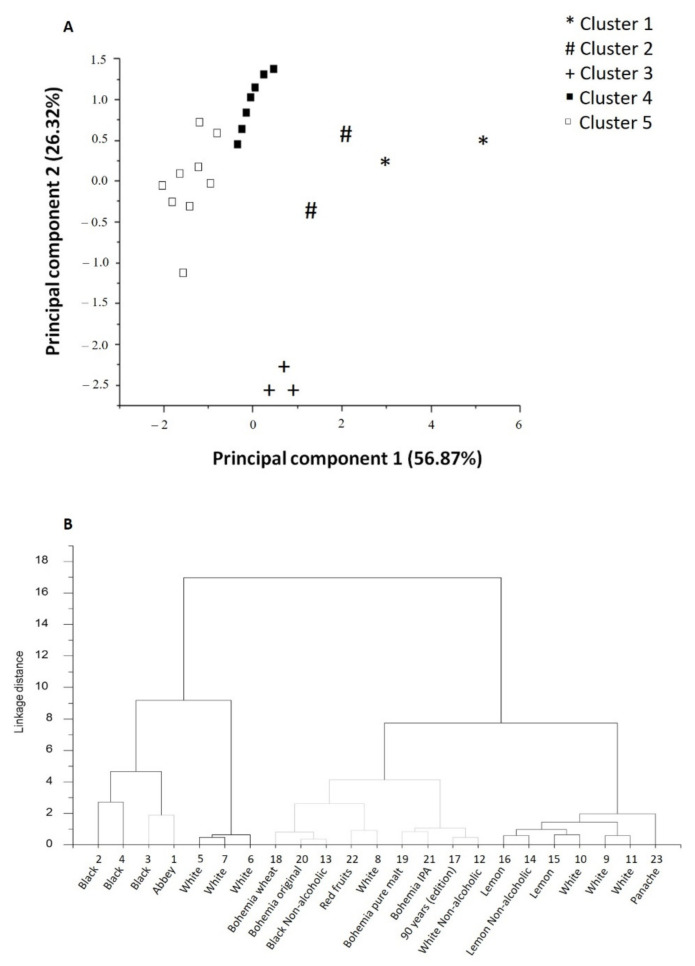
(**A**) Grouping resulting from the analysis of clusters. Different colors refer to different groups. The distribution of the samples is represented according to the values for CPs 1 and 2; (**B**) Dendrogram for beers obtained by hierarchical group analysis.

**Table 1 foods-10-01144-t001:** Characteristics of Portuguese beers of different types and styles analyzed in this study (*n* = 23).

Sample No.	Beer Style	Type	Color	Ethanol (% *v*/*v*) ^a^	Ingredients ^a^
1	Abbey	Ale	Brown	6.4	Water, barley malt, unmalted cereals (barley), sugar, hops.
2	Black	Ale	Brown	5.0	Water, barley malt, unmalted cereals (barley), sugar, coloring: E150c, hops.
3	Black	Lager	Brown	4.2	Water, barley malt, unmalted cereals (corn and barley), sugar, coloring: E150c, glucose syrup, hops
4	Black	Lager	Pale	4.1	Water, malt (barley), unmalted cereals (corn or rice), hops
5	White	Lager	Pale	5.2	Water, barley malt, unmalted cereals (maize and barley), hops.
6	White	Lager	Pale	5.1	Water, barley malt, unmalted cereals (maize and barley), hops.
7	White	Lager	Pale	5.0	Water, barley malt, corn, barley and hops. Contains barley
8	White	Lager	Pale	4.8	NI
9	White	Lager	Pale	5.0	NI
10	White	Lager	Pale	4.6	NI
11	White	Lager	Pale	5.0	Water, barley malt, corn, hops, barley, dye: E150c, stabilizer: E405.
12	White Non-alcoholic	Lager	Pale	0.5	Water, barley malt, unmalted cereals (corn and barley), flavorings, hops.
13	Black Non-alcoholic	Lager	Brown	0.5	Water, barley malt, unmalted cereals (maize and barley), carbon dioxide, dye: E150 C, hops.
14	Lemon Non-alcoholic	Lager	Pale	0.0	50% non-alcoholic beer (approx. 0.03%) (water, barley malt, carbon dioxide and hops), 39.5% carbonated water, fructose, 4.9% fruit juices based on concentrate (2, 7% lemon; 1.4% orange; 0.5% lime; 0.3% acerola), concentrated lemon extract, natural aromas, stabilizer: locust bean gum
15	Lemon	Lager	Pale	2.0	Water, beer (44%) (water, barley malt, unmalted cereals (corn and barley) and hops), fructose, concentrate-based fruit juice (lemon 1.9% orange 1.8%), stabilizer: pectin, natural flavors, acidifying: citric acid, barley malt extract.
16	Lemon	Lager	Pale	2.0	Carbonated water, beer (water, malt (barley), unmalted cereals (corn and barley), hops), fructose, concentrate fruit juices (lemon 2.7%, orange 1.4%, lime 0.5%, acerola 0.3%), concentrated lemon extract, natural aromas, stabilizer: locust bean gum.
17	90 years (edition)	Lager	Pale	5.0	Water, malt and special hops (Styrian, Tettnang and Saaz).
18	Bohemia wheat	Lager	Pale	5.5	Water, barley malt, wheat malt, hops, natural aromas.
19	Bohemia pure malt	Lager	Pale	6.0	Water, barley malt, corn, barley, coloring: caramel and hops.
20	Bohemia original	Lager	Pale	6.2	Water, barley malt, corn, barley, coloring: caramel and hops. Contains barley
21	Bohemia IPA	Ale	Pale	7.0	NI
22	Red fruits	Lager	Pale	2.2	Carbonated water, beer (water, barley malt, corn, barley and hops extract), fructose syrup, concentrate-based fruit juices (lime 1.7%, lemon 1.6%, blueberry 0.05% and 0.05% raspberry), purple carrot concentrate, natural aroma, barley malt extract, concentrated lemon extract, stabilizer: locust bean gum. Contains barley.
23	Panache	Lager	Pale	0.0	50% non-alcoholic beer (water, barley malt, corn, barley, glucose syrup, hops, color: E150c, stabilizer: E405); Flavored soda. Contains 50% sweeteners (carbonated water, E330 acidifier, aroma, sweeteners: acesulfame K, sucralose).

^a^ As reported in the label; NI—No Information on the label.

**Table 2 foods-10-01144-t002:** Phenolic content and antioxidant capacity of Portuguese beers.

		Phenolic Content			Antioxidant Capacity	
Beer samples		Total phenols (g GA L^−1^)	Ortho-diphenols (g GA L^−1^)	Flavonoids (g CAT L^−1^)	ABTS (mmol trolox L^−1^)	DPPH (mmol trolox L^−1^)
1	Abbey	^X^ 0.706 ± 0.027 ^gh^	0.401 ± 0.015 ^efz^	0.085 ± 0.014 ^abcdef^	0.049 ± 0.001 ^d^	0.021 ± 0.001 ^a^
2	Black	0.824 ± 0.074 ^h^	1.552 ± 0.142 ^h^	0.151 ± 0.017 ^f^	0.102 ± 0.007 ^e^	0.019 ± 0.002 ^a^
3	Black	0.506 ± 0.065 ^ef^	0.704 ± 0.007 ^g^	0.098 ± 0.016 ^abcdef^	0.016 ± 0.001 ^b^	0.020 ± 0.001 ^a^
4	Black	0.515 ± 0.034 ^ef^	0.718 ± 0.015 ^ef^	0.109 ± 0.009 ^bcdef^	0.107 ± 0.003 ^e^	0.021 ± 0.001 ^a^
5	White	0.362 ± 0.027 ^ef^	1.634 ± 0.044 ^h^	0.041 ± 0.003 ^abcd^	0.023 ± 0.001 ^c^	0.023 ± 0.001 ^a^
6	White	0.269 ± 0.051 ^ab^	1.561 ± 0.011 ^h^	0.033 ± 0.002 ^ab^	0.021 ± 0.001 ^bc^	0.023 ± 0.001 ^a^
7	White	0.304 ± 0.048 ^abcd^	1.801 ± 0.087 ^i^	0.043 ± 0.005 ^abcde^	0.023 ± 0.001 ^c^	0.022 ± 0.001 ^a^
8	White	0.278 ± 0.029 ^abc^	0.074 ± 0.024 ^a^	0.076 ± 0.026 ^abcdef^	0.019 ± 0.000 ^bc^	0.037 ± 0.001 ^de^
9	White	0.354 ± 0.036 ^bcde^	0.170 ± 0.011 ^abc^	0.042 ± 0.028 ^abcd^	0.020 ± 0.000 ^bc^	0.032 ± 0.002 ^bc^
10	White	0.276 ± 0.034 ^abc^	0.185 ± 0.028 ^abc^	0.039 ± 0.006 ^abc^	0.017 ± 0.001 ^b^	0.038 ± 0.001 e
11	White	0.396 ± 0.012 ^bcde^	0.122 ± 0.009 ^ab^	0.041 ± 0.017 ^abcd^	0.009 ± 0.002 ^a^	0.034 ± 0.003 ^cde^
12	White Non-alcoholic	0.485 ± 0.078 ^def^	0.082 ± 0.063 ^a^	0.045 ± 0.010 ^abcde^	0.020 ± 0.002 ^bc^	0.034 ± 0.001 ^cde^
13	Black Non-alcoholic	0.461 ± 0.012 ^cdef^	0.419 ± 0.046 ^f^	0.125 ± 0.022 ^ef^	0.018 ± 0.001 ^bc^	0.036 ± 0.001 ^cde^
14	Lemon Non-alcoholic	0.267 ± 0.030 ^ab^	0.188 ± 0.008 ^abc^	0.042 ± 0.025 ^abcd^	0.009 ± 0.001 ^a^	0.032 ± 0.003 ^bc^
15	Lemon	0.304 ± 0.022 ^abcd^	0.099 ± 0.007 ^ab^	0.020 ± 0.005 ^a^	0.008 ± 0.001 ^a^	0.037 ± 0.000 ^e^
16	Lemon	0.240 ± 0.022 ^ab^	0.163 ± 0.010 ^abc^	0.032 ± 0.016 ^ab^	0.008 ± 0.001 ^a^	0.035 ± 0.001 ^cde^
17	90 years (edition)	0.500 ± 0.172 ^ef^	0.154 ± 0.020 ^abc^	0.060 ± 0.012 ^abcde^	0.016 ± 0.002 ^b^	0.033 ± 0.002 ^cd^
18	Bohemia wheat	0.397 ± 0.099 ^bcde^	0.199 ± 0.024 ^abc^	0.116 ± 0.042 ^cdef^	0.021 ± 0.001 ^bc^	0.034 ± 0.003 ^cde^
19	Bohemia pure malt	0.607 ± 0.067 ^fg^	0.281 ± 0.012 ^cde^	0.067 ± 0.005 ^abcde^	0.018 ± 0.001 ^bc^	0.035 ± 0.001 ^cde^
20	Bohemia original	0.500 ± 0.048 ^bcde^	0.357 ± 0.050 ^def^	0.123 ± 0.093 ^def^	0.018 ± 0.001 ^bc^	0.034 ± 0.000 ^cde^
21	Bohemia IPA	0.487 ± 0.051 ^def^	0.228 ± 0.012 ^bcd^	0.078 ± 0.028 ^abcdef^	0.018 ± 0.001 ^bc^	0.035 ± 0.001 ^cde^
22	Red fruits	0.312 ± 0.054 ^abcd^	0.451 ± 0.026 ^f^	0.089 ± 0.020 ^abcdef^	0.020 ± 0.001 ^bc^	0.034 ± 0.000 ^cde^
23	Panache	0.153 ± 0.013 ^a^	0.191 ± 0.012 ^abc^	0.023 ± 0.012 ^a^	0.010 ± 0.001 ^a^	0.028 ± 0.003 ^b^
*p*-value		^Y^ ***	***	***	***	***

^X^ Values are presented with mean ± SD (*n* = 3). Different letters indicate significantly different results (ANOVA. *p* > 0.05). according to the Tukey test. ^Y^ Significance: not significant. N.S. (*p* > 0.05); *** significant with *p* < 0.001.

**Table 3 foods-10-01144-t003:** Phenolic composition of Portuguese beers.

						**Phenolic Acids**													
Phenolic compounds (ug/mL) Samples					Gallic acid			Protocatechuic acid		Hydroxybenzoic acid			Vanillic acid			Caffeic acid	Ferulic acid		Syringic acid
1		Abbey			^X^ 95.80 ± 3.71			-		-			-			-	-		10.36 ± 1.150
2		Black			24.37 ± 0.41			28.43 ± 0.26		12.33 ± 0.25			-			-	1.49 ± 0.06		-
3		Black			70.43 ± 6.51			-		-			-			-	-		6.69 ± 0.13
4		Black			68.62 ± 2.90			-		-			-			-	-		8.38 ± 0.18
5		White			-			3.51 ± 0.56		-			-			-	-		5.56 ± 0.58
6		White			52.64 ± 3.09			-		-			-			-	-		11.67 ± 0.06
7		White			48.12 ± 7.13			-		-			-			-	-		9.25 ± 0.63
8		White			45.03 ± 1.27			-		-			-			-	-		7.24 ± 0.15
9		White			67.77 ± 3.89			-		-			-			-	-		11.33 ± 0.94
10		White			46.46 ± 3.09			-		-			-			-	-		6.81 ± 0.74
11		White			45.07 ± 0.05			-		-			-			-	-		5.89 ± 0.60
12		White Non-alcoholic			79.76 ± 11.19			-		-			-			-	-		12.40 ± 1.62
13		Black Non-alcoholic			76.93 ± 7.42			-		-			-			-	-		-
14		Lemon Non-alcoholic			25.08 ± 0.35			-		-			-			-	-		-
15		Lemon			35.76 ± 3.47			-		-			-			-	0.31 ± 0.00		4.05 ± 0.43
16		Lemon			25.01 ± 0.02			-		-			-			-	-		2.87 ± 0.01
17		90 years (edition)			72.67 ± 5.50			-		-			-			-	-		3.26 ± 0.10
18		Bohemia wheat			64.62 ± 1.91			-		-			1.33 ± 0.09			-	-		3.47 ± 0.28
19		Bohemia pure malt			80.27 ± 12.68			-		-			-			-	-		3.72 ± 0.39
20		Bohemia original			60.87 ± 5.14			-		-			-			-	-		7.54 ± 0.50
21		Bohemia IPA			71.70 ± 1.33			-		-			-			-	-		2.56 ± 0.13
22		Red fruits			46.65 ± 3.96			-		-			-			29.24 ± 3.638	-		3.20 ± 0.42
23		Panache			29.21 ± 4.11			-		-			-			-	-		2.28 ± 0.31
			**Flavonoids**																
Phenolic compounds (ug/mL) Samples			Kaempferol	Catechin			Kaempferol-3-*O*-(malonyl)glucoside		Epicatechin		Kaempferol-3-*O*-(6”-*O*-manonyl)glucoside)	Quercetin		Rutin (Quercetin-3-*O-*rutinoside)	Kaempferol-3-*O*-xylosylglucoside		Kaempferol-3-*O*-glucoside	Delphinidin-3-*O*-glucoside	Gallocatechin
1	Abbey		^X^ 0.20 ± 0.03	40.82 ± 5.26			-		215.10 ± 22.58		-	-		-	-		-	-	63.59 ± 7.84
2	Black		0.21 ± 0.01	86.47 ± 5.31			-		-		-	-		-	-		-	-	-
3	Black		0.23 ± 0.02	18.91 ± 0.40			-		93.32 ± 8.559		-	-		-	-		-	-	95.23 ± 10.25
4	Black		0.17 ± 0.02	31.49 ± 0.27			-		-		-	-		-	-		-	-	66.50 ± 2.40
5	White		-	96.13 ± 17.00			-		-		-	-		-	-		-	-	21.44 ± 2.87
6	White		-	78.97 ± 1.30			-		-		-	-		-	-		-	-	58.57 ± 0.00
7	White		0.09 ± 0.00	2.27 ± 0.10			-		28.45 ± 3.15		-	-		-	-		-	-	29.94 ± 2.19
8	White		-	1.41 ± 0.01			-		43.48 ± 1.84		-	-		-	-		-	-	27.19 ± 1.47
9	White		-	2.30 ± 0.21			-		87.71 ± 3.84		-	-		-	-		-	-	86.70 ± 1.89
10	White		0.05 ± 0.00	3.37 ± 0.40			-		45.47 ± 6.30		-	-		-	-		-	-	23.88 ± 1.03
11	White		-	-			-		64.89 ± 0.12		-	-		-	-		-	-	37.76 ± 0.40
12	White Non-alcoholic		-	-			-		144.20 ± 20.56		-	-		-	-		-	-	89.63 ± 2.01
13	Black Non-alcoholic		-	3.87 ± 0.00			-		86.26 ± 2.47		-	-		-	-		-	-	76.18 ± 0.23
14	Lemon Non-alcoholic		-	-			0.31 ± 0.02		40.11 ± 3.52		0.34 ± 0.01	0.66 ± 0.03		2.00 ± 0.20	1.67 ± 0.12		0.21 ± 0.00	-	144.00 ± 10.93
15	Lemon		-	44.10 ± 5.42			-		3.28 ± 0.18		-	-		-	-		-	-	83.50 ± 2.00
16	Lemon		-	-			0.391 ± 0.01		7.94 ± 0.48		0.44 ± 0.01	0.73 ± 0.03		0.46 ± 0.02	1.73 ± 0.08		0.27 ± 0.03	-	73.89 ± 0.58
17	90 years (edition)		0.13 ± 0.02	-			1.07 ± 0.11		197.50 ± 18.23		-	-		-	-		-	-	74.00 ± 1.37
18	Bohemia wheat		-	-			-		221.50 ± 10.91		-	-		-	-		-	-	53.95 ± 1.26
19	Bohemia pure malt		0.15 ± 0.01	155.30 ± 2.10			-		-		-	-		-	-		-	-	43.88 ± 0.76
20	Bohemia original		0.14 ± 0.02	-			-		35.61 ± 2.88		-	-		-	-		-	-	47.87 ± 4.05
21	Bohemia IPA		-	2.60 ± 0.11			-		158.50 ± 3.16		-	-		-	-		-	-	48.74 ± 2.26
22	Red fruits		0.06 ± 0.01	6.81 ± 0.88			0.24 ± 0.01		11.34 ± 1.12		0.44 ± 0.07	0.27 ± 0.04		-	-		-	a	137.80 ± 7.12
23	Panache		0.06 ± 0.04	-			-		21.08 ± 1.82		-	-		-	-		-	-	72.56 ± 7.40

^X^ Data presented as mean ± SD (*n* = 3). (a) An anthocyanidin has been identified, namely a delphinidin.

**Table 4 foods-10-01144-t004:** Pearson’s correlation coefficient test (r) and weight of each variable for the main components.

	Pearson’s Correlation Coefficient					Weight	
	*Ortho*-Diphenols	Total Phenols	Flavonoids	ABTS	DPPH	PC1	PC2
*Ortho*-diphenols	1	0.15627	0.10443	0.4078 ***	−0.75267 ***	0.45471	0.41383
Total phenols	0.15627	1	0.61357 ***	0.60149 ***	−0.33056 **	0.3765	−0.59563
Flavonoids	0.10443	0.61357 ***	1	0.50369 ***	−0.17144	0.41221	0.49995
ABTS	0.4078 ***	0.60149 ***	0.50369 ***	1	−0.58414 ***	0.51781	0.08682
DPPH	−0.75267 ***	−0.33056 **	−0.17144	−0.58414 ***	1	−0.4620	0.4658

* refers the correlation between samples (*p* ≤ 0.05). ** value < 0.01; *** value < 0.001.

## Data Availability

Not applicable.

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
