# Peer review of "Characterization and Discrimination of Commercial Portuguese Beers Based on Phenolic Composition and Antioxidant Capacity"

_foods, 2021, doi:10.3390/foods10051144_

Round 1

Reviewer 1 Report

The title of manuscript is “ Evaluation of phytochemical composition of Portuguese beers for rationale marketable selection”. The objectives in this research were to study and to characterize some of the most common Portuguese beers available on the market, by the assessment of their content in phenolics and antioxidant capacity.

Line numbers should be on the right side of the text so that they do not conflict with the text on the first page.

The Introduction of the study is brief and provides with some general information about the studied problems.

Line 50: The Celsius degree should be without underline.

The Materials and Methods section provides the reader with not enough information to repeat the experiments conducted. The strength of the work are the use of advanced statistical methods to analyze the obtained results but numbered results should be replaced with the corresponding sample names. This will make the charts readable and easy to analyze for the reader. I have a series of questions about the Principal Component Analysis (PCA) used in the work:

On the basis of which criterion was the optimal number of main components obtained in the PCA analysis determined?

How many columns and rows had a data matrix for PCA?

Is the input matrix automatically scaled?

Please put this information in this chapter.

In the Results and discussion chapter contains information should be supplemented on discussing with the other items from the last years of publication including similar problems.

The conclusions are well and were supported by the data.

The literature used is appropriate but should be supplementing about the items from the last years of publication. 

Author Response

Dear Editor of FOODS,

In reply to the review performed to the paper entitled "Evaluation of phytochemical composition of Portuguese beers for rationale marketable selection", we would like to acknowledge the valuable comments performed by the editor that kindly accepted to revise our manuscript.

We would like to confirm that we have addressed most issues and answered the questions that have been made.

We hope the answers below and modifications introduced in the manuscript are clear and concise enough as required by the reviewers, in order to enable the publication of the manuscript in FOODS.

Answer to referee’s comments and queries

Detailed responses to Reviewer 1

Reviewer´s comment: Line numbers should be on the right side of the text so that they do not conflict with the text on the first page.

Our reply:

We thank the reviewer suggestion. We revised the manuscript according to the reviewer comments.

Reviewer´s comment: The Introduction of the study is brief and provides with some general information about the studied problems.

Our reply: We appreciate the reviewer comment. The Introduction has been improved in order to provide more information about recent studies related to this topic.

Reviewer´s comment: Line 50: The Celsius degree should be without underline.

Our reply: We thank the reviewer comment. We removed the underline.

Reviewer´s comment: The Materials and Methods section provides the reader with not enough information to repeat the experiments conducted. The strength of the work are the use of advanced statistical methods to analyze the obtained results but numbered results should be replaced with the corresponding sample names. This will make the charts readable and easy to analyze for the reader.

Our reply: We thank the reviewer suggestion. The Material and Methods section concerning the statistical analysis was improved. Since the brand name of the beer samples cannot be used, the authors decided to introduce the reference names in the figures: 1 – Abbey, 2 – Black, …, 23 – Panache.

Reviewer´s comment: I have a series of questions about the Principal Component Analysis (PCA) used in the work:

On the basis of which criterion was the optimal number of main components obtained in the PCA analysis determined?

Our reply: We appreciate the reviewer comment. The criterion used to decide the number of principal components (PCs) to take in the case of PCA algorithm, is the one that explain the maximum amount of variance in the dataset. This information was included in the manuscript, at the statistical section of Material and Methods.

Reviewer´s comment: How many columns and rows had a data matrix for PCA?

Our reply: We thank the reviewer comment. The columns, represented by the variables (ABTS, total phenols, etc) were 5, and the rows (observations: beers analysed) were 23. This information was also included in the manuscript, at the statistical section of Material and Methods.

Reviewer´s comment: Is the input matrix automatically scaled? Please put this information in this chapter.

Our reply: We thank the reviewer comment. The data were manually normalized to 0–1 range, taking into account the maximum value obtained from each assay. This information has been added to the manuscript.

Reviewer´s comment: In the Results and discussion chapter contains information should be supplemented on discussing with the other items from the last years of publication including similar problems.

Our reply: We appreciate the reviewer comment. The results and discussion section has been improved in order to provide more information about recent studies related to this topic.

Reviewer´s comment: The conclusions are well and were supported by the data.

Our reply: We thank the reviewer comment.

Reviewer´s comment: The literature used is appropriate but should be supplementing about the items from the last years of publication. 

Our reply: We appreciate the reviewer comment. The manuscript has been improved in order to provide more information about recent studies related to this topic.

The manuscript was modified accordingly to the reviewer’s suggestions.

If you need any further assistance or information, please, do not hesitate to contact us.

Sincerely,

                                                                                   Ana Novo Barros

Reviewer 2 Report

This study evaluated the phytochemical composition and antioxidant capacity of commonly consumed beers in Portugal to characterize them. Although it provides some interesting scientific results, several critical limitations to the current study exist. Notably, a strong rationale for the importance of the study was not offered; and hypotheses were not stated and therefore not clearly tested. 

  • The title must be revised to state a major finding of the study rather than what was done.
  • There is no section 4 (Discussion).
  • The results are good, but the discussion needs to be improved, and more and updated references must be used.
  • The last sentence in the conclusion section (L 454-456) does not appear to be supported by the data obtained in this study.
  • Other minor comments are;
    • Add references to all methods.
    • Move the repeated sentences (L 132-133, L-147) to the 2.6 statistical analysis section.
    • Line 144, 145: Do not need to list City and Country for the devices re-introduced in the manuscript.
    • Use colored figures (especially Figure 2A) or change the explanations in L 397-403.

Author Response

Dear Editor of FOODS,

In reply to the review performed to the paper entitled "Evaluation of phytochemical composition of Portuguese beers for rationale marketable selection", we would like to acknowledge the valuable comments performed by the editor that kindly accepted to revise our manuscript.

We would like to confirm that we have addressed most issues and answered the questions that have been made.

We hope the answers below and modifications introduced in the manuscript are clear and concise enough as required by the reviewers, in order to enable the publication of the manuscript in FOODS.

Answer to referee’s comments and queries

Detailed responses to Reviewer 2

Reviewer´s comment:  This study evaluated the phytochemical composition and antioxidant capacity of commonly consumed beers in Portugal to characterize them. Although it provides some interesting scientific results, several critical limitations to the current study exist. Notably, a strong rationale for the importance of the study was not offered; and hypotheses were not stated and therefore not clearly tested. 

Our reply: We thank the reviewer comment. It is well known about the rich constitution on phenolic compounds and, consequently, the high antioxidant capacity revealed by beers, being responsible for some biological properties. In fact, a growing interest has been focused on beer, due to its phenolic antioxidant component coupled with low ethanol content. Many studies have been conducted on beer phenolics and their antioxidant activities, however, to the best of our knowledge, no studies about the phenolic composition were made in industrial Portuguese beers.      Thus, the originality of this work is related to the study of the major phenolic compounds and derivatives of twenty-three Portuguese commercial beers using HPLC-DAD and other spectrophotometric methods. In addition, the antioxidant capacities of all beer extracts were studied using in vitro screening assays. Additionally, a deep statistical analysis was conducted through a Principal Component Analysis, a Pearson correlation test, and a dendogram to evaluate the regularity and differences among samples, namely to distinguish differences in the phenolic content and antioxidant capacity of beer samples according to their styles.       

Thus, the intention of this research was to give an overview of polyphenolic compounds in beers which can be found on the Portuguese market. Since no similar research has been conducted in Portugal, this survey gives an important insight in different types and beer styles found in supermarkets.

This information has been added into the manuscript.

Reviewer´s comment: The title must be revised to state a major finding of the study rather than what was done.

Our reply: We kindly thank the reviewer attention. We suggest to change the title to: “Characterization and discrimination of commercial Portuguese beers based on phenolic composition and antioxidant capacity”.

Reviewer´s comment: There is no section 4 (Discussion).

Our reply: We thank the reviewer comment. The authors choose to have results and discussion as one section. However, by lapse, the “discussion” mention was not included in the manuscript. Now, it has been included.

Reviewer´s comment: The results are good, but the discussion needs to be improved, and more and updated references must be used.

Our reply: We appreciate the reviewer comment. The Results and Discussion section has been improved in order to provide more information about recent studies related to this topic.

Reviewer´s comment: The last sentence in the conclusion section (L 454-456) does not appear to be supported by the data obtained in this study.

Our reply: We appreciate the reviewer attention. We removed the sentence.

Reviewer´s comment: Other minor comments are;

Add references to all methods.

Our reply: We thank the reviewer suggestion. All the methods have been accordingly referenced.

Reviewer´s comment: Move the repeated sentences (L 132-133, L-147) to the 2.6 statistical analysis section.

Our reply: We kindly thank the reviewer attention. The sentences were moved to the 2.6 statistical section.

Reviewer´s comment: Line 144, 145: Do not need to list City and Country for the devices re-introduced in the manuscript.

Our reply: We thank the reviewer attention. We removed this information.

Reviewer´s comment: Use colored figures (especially Figure 2A) or change the explanations in L 397-403.

Our reply: We kindly thank the reviewer attention. The sentences were reformulated in order to avoid the figure colors.

The manuscript was modified accordingly to the reviewer’s suggestions.

If you need any further assistance or information, please, do not hesitate to contact us.

Sincerely,

                                                                                   Ana Novo Barros

Round 2

Reviewer 2 Report

No further comment.